# Improving Coconut Using Modern Breeding Technologies: Challenges and Opportunities

**DOI:** 10.3390/plants11243414

**Published:** 2022-12-07

**Authors:** Thayalan Arumugam, Muhammad Asyraf Md Hatta

**Affiliations:** Department of Agriculture Technology, Faculty of Agriculture, Universiti Putra Malaysia, Serdang 43400, Selangor, Malaysia

**Keywords:** coconut, precision phenotyping, genomic selection, genome editing, speed breeding

## Abstract

Coconut (*Cocos nucifera* L.) is a perennial palm with a wide range of distribution across tropical islands and coastlines. Multitude use of coconut by nature is important in the socio-economic fabric framework among rural smallholders in producing countries. It is a major source of income for 30 million farmers, while 60 million households rely on the coconut industry directly as farm workers and indirectly through the distribution, marketing, and processing of coconut and coconut-based products. Stagnant production, inadequate planting materials, the effects of climate change, as well as pests and diseases are among the key issues that need to be urgently addressed in the global coconut industry. Biotechnology has revolutionized conventional breeding approaches in creating genetic variation for trait improvement in a shorter period of time. In this review, we highlighted the challenges of current breeding strategies and the potential of biotechnological approaches, such as genomic-assisted breeding, next-generation sequencing (NGS)-based genotyping and genome editing tools in improving the coconut. Also, combining these technologies with high-throughput phenotyping approaches and speed breeding could speed up the rate of genetic gain in coconut breeding to solve problems that have been plaguing the industry for decades.

## 1. Introduction

Coconut (*Cocos nucifera* L.) is a perennial palm with 32 (2n = 16) chromosomes that grows on tropical islands and coastlines [1]. Previous genetic studies using molecular markers have divided the existing coconut palm genotypes into two major groups based on origin and domestication, the Pacific-Southeast Asia group, which has many variations, and the Indo-Atlantic group, which contains fewer variations [2]. Furthermore, the palm is subdivided into two morphological types: tall and dwarf. Tall is cross-pollinating, heterozygous, has a long lifespan and matures slowly. Dwarf cultivar, on the other hand, is self-pollinating, homozygous and mature faster than tall cultivar [3]. Dwarfs are typically found near human settlements and contain traits that reflect human selection, such as slow trunk growth and self-pollination. This is in contrast to tall types, which lack self-pollination and domesticated characteristics [4].

Since coconut has various economical uses and every part of the palm is useful to humans, it is often regarded as a ‘tree of life’ [5]. Coconut’s contributions to the food and non-food chains have a significant impact on the socioeconomic well-being of vast rural populations in tropical countries. In the international coconut community member countries, the number of households classified as growers is estimated to be 22,738,000 [6]. It is a major source of income for many people living in rural areas around the world. Thirty million people work as farmers, and 60 million households rely on the coconut industry, both directly as farm workers and indirectly as distributors, marketers, and processors of coconut and coconut-based products [6].

The current state of worldwide coconut planted area and total nut output indicates that a significant amount of replanting is required. This is because little or no replanting activities have been recorded for the past 20 to 30 years [6], and on the farms, senile coconut trees that are over 60 years old make over half of the available coconut trees. Besides, there is evidence of potential yield gap with actual yield cultivated around the globe [7]. According to previous studies, factors such as low-quality planting material, poor agronomic practices, climatic stresses, and biotic factors contribute to the gap in estimated potential yield and actual output which is between 33% and 84% for nuts and copra [3]. To address these issues, the development of superior varieties/hybrids with high yield, resistance to pathogens and resilient to climate change is essential, in addition to effective management and cultural practices. Although conventional crossings have been utilized to achieve these coconut breeding goals, it often takes a long time due to the biological nature of the coconut, which includes large stature, lengthy generation, recalcitrant seeds, and a low multiplication rate [8]. The advent of new biotechnology tools has transformed the way breeders create and exploit the genetic variations of crops. 

This review highlights the challenges of current breeding strategies and the potential of biotechnological approaches in improving coconut productivity. We will begin with a brief review of the importance of conserving and managing germplasm as a valuable source of novel genetic variations. Genotyping approaches based on next-generation sequencing generate markers for genome analysis with a reduced degree of complexity. Combined with the precision phenotyping of large field-based data, this would enable breeders to more efficiently link genetic variations with the relevant phenotypes. Next-generation sequencing platforms would further facilitate this process through the generation of genomic resources including high-quality reference genomes, transcriptomes, and pangenomes. Following the identification of candidate genes associated with the desired trait, functional characterization can be carried out using genome editing tools. Furthermore, the potential of adopting speed breeding protocol to shorten the coconut generation time will be discussed. To complement the replanting efforts, superior coconut palms that are more resistant to biotic and abiotic stress as well as higher quality can be mass propagated using tissue culture techniques. The recent advances and potential strategies discussed throughout this review ultimately aim to accelerate coconut breeding program, enabling the breeders to overcome challenges that have been plaguing the coconut improvement for decades. 

## 2. Collection, Conservation, and Utilization of Coconut Genetic Resources

Germplasm collections are genetic resources that underpin a strong crop breeding program. Long-term natural evolution, artificial selection, or domestication processes can all result in genetic variations within a crop. In coconut, although it is a monotypic species, significant variation can still be observed in the existing populations in terms of height, leaf morphology, nut characteristics, resistance to pest and disease, among other traits, which have been utilized for evaluation of coconut germplasm [9]. However, land tenure issues, urbanization, crop shifts, and natural and human-made calamities have narrowed the genetic base of coconut, posing a threat to its genetic diversity. A global loss of 54 cultivars, which represents 13% of the existing global collections was reported due to land tenure problems. The most recent dispute occurred between villagers and international genebank in Côte d’Ivoire, for the replanting of eight hectares of coconut palms [10]. Thus, it is important to conserve these genetic resources to facilitate their use for developing improved varieties/hybrids. 

The effort to collect and conserve the natural diversity of coconut germplasm has been led by the International Coconut Genetic Resources Network (COGENT), an association comprised of 39 coconut-growing countries worldwide. Since its establishment in 1992, COGENT has gathered 1760 accessions which are maintained as ex situ collections in 23 COGENT member-countries, for a total of 25 field genebanks [11]. All the institutes located in those countries are given a mandate by their respective governments to conserve, characterize, and coordinate the germplasm exchange among key stakeholders [10].

In order to promote a more efficient and effective system of germplasm conservation, research, and sharing activities, a multi-site International Coconut Genebank (ICG) has been established by the COGENT in five regional genebanks; Brazil, Côte d’Ivoire, India, Indonesia, and Papua New Guinea [10,12]. In 2010, a catalogue of conserved coconut germplasm was released, which provides comprehensive information including illustrations and detailed information on 116 conserved accessions from 27 coconut-growing countries [12]. The inclusion of more accessions from diverse geographic regions into this open-source dataset will add novel population-based collections to target hidden adaptive variation in coconut breeding [13]. 

Most of the earlier studies on germplasm characterization and evaluation have focused on morphological and agronomical traits [14,15,16]. This approach requires extensive observation of plants in the field such as duration to planting and flowering, fruit and bunch production, fruit component analysis, and pest and disease resistance. Since coconut is a polymorphous plant, its morphological traits vary considerably depending on the environmental variables including soil, climate, and time of year [10]. This may create significant variation in the data collection between the different ICGs and lead to misidentification of palms and errors within a genebank. To avoid this problem, molecular markers such as simple sequence repeats (SSR) [17,18,19], random amplified polymorphic DNA (RAPD) [20], restriction fragment length polymorphisms (RFLP) [21], and amplified fragment length polymorphisms (AFLP) [22] have been utilized in the characterization and evaluation process. Although this approach is more expensive and requires specialized facilitates, it offers greater accuracy in determining the genetic diversity of ex situ coconut germplasm at DNA level to accelerate the development of the elite varieties for large-scale coconut production.

The genotypically characterized germplasm can be further studied for the signatures of local adaptation, especially to abiotic factors using current genomic analytical tools coupled with environmental variables, such as genome-wide selection scans (GWSS) [23] and genome-environment associations (GEA) [24]. By identifying locally adapted genotypes to climate change, the genomic basis underlying the important traits like drought and heat tolerance can be explored and utilized for pre-breeding and breeding of climate-resilient coconut palms. 

To uncover the genes that underlie the important traits from the screening of ex situ and in situ coconut germplasm collections, various modern breeding approaches could be utilized such as high-throughput genotyping based on next-generation sequencing, precision phenotyping, and available genomic resources. Genomic-assisted breeding, marker-assisted breeding, and genome editing all can be employed to transfer useful genetic variation from naturally adapted genotypes identified from the germplasm. 

## 3. Coconut Breeding’s Goals, Accomplishments, and Prospects

### 3.1. High Yielding Coconut

About 8000 years ago, the selection and domestication of coconuts for uncontaminated water sources [25] changed in favor of commercialization, with yield performance becoming the key criterion for selection and domestication. Extensive studies on the mass selection of mother palms based on desirable characteristics, as well as progeny trials in coconut-producing countries, have resulted in the development of seed coconut palm selection criteria. For many coconut breeders and farmers, high yield is the most important breeding goal. However, it remains unclear whether the quantity of nuts or the size of the nuts (copra weight) would be used to estimate yield. Since there is a negative relationship between the number of nuts and the weight of the copra [3] it would be difficult to increase both the number of nuts and their size through selection alone. 

Inter-varietal crossing, such as dwarf and tall, has resulted in hybrids with a higher quantity of nuts but a lower yield of copra. [26] has reported that hybrids between dwarf and tall showed increased nut yield of more than 45% relative to the tall variety. High nut production will compensate for the low copra yield. Dwarf x tall hybrids can produce coconut goods with highly desirable characteristics including early bearing, large fruit, and copra yield [27]. A hybrid called MAWA (Malayan Dwarf x West African Tall) that was introduced in the middle of the 1970s to replace the tall variety experienced a fall in production because of competition from other oil crops like oil palm and cocoa pod borer infestation [28]. Later, a program in Malaysia focused on breeding dwarf-tall hybrids designed to take advantage of the tall trees’ enormous nuts and the dwarfs’ high nut production, in which MATAG (Malayan Dwarf x Tagnanan Tall) [28] was introduced. The popularity of MATAG among farmers increased due to the 18% increased copra yield, ease of dehusking for dry nuts, large nut size with thinner husk and high-water content which shows superiority of MATAG over MAWA [29]. Meanwhile, a high number of copra yield is attained via crossing tall to tall variety and reported to have low nut production and is not promising in terms of early flowering [30]. Kapruwana, a hybrid resulting from San Ramon and Sri Lankan Green Dwarf (SLGD) is a breakthrough because it combines the large fruit trait from San Ramon and the high number of fruits trait from SLGD, giving rise to hybrids that bear a high number of fruits along with increased copra yield [3].

Despite all these achievements, it is still not practical to produce high-performing hybrid seeds on a large scale. This is because production of hybrid seed is time consuming and requires wide seedbeds with sufficient pollen for assisted pollination, which leads to a hike in total cost of producing one hybrid seed [31]. Therefore, coconut breeding should focus on large-scale high-quality seedling production in a shorter period, which will speed up the replanting process. Surprisingly, this issue is not well addressed, and there is little evidence of research being conducted to establish a technique or protocol capable of producing high-quality F1 hybrid seedlings in a timely manner.

More studies using multi-omics and utilization of recent advancements in biotechnological tools should enhance and simplify the selection of varieties with desirable traits including resistance towards biotic and abiotic challenges. This will ensure the coconut industry rise in the era of molecular breeding.

### 3.2. Climate-Smart Coconut

Weather unpredictability has a significant impact on crop growth, development, and yield. Heat stress will affect the reproductive development and cause a decline in the productivity of plants [32]. Abiotic factors such as rainfall, day/night temperature regimes, relative humidity, sunshine length, and vapor pressure deficit all appear to have a significant impact on agricultural output [33]. Critical stages of plant growth including pollination, flowering, and fruit development can be affected by drought and excessive temperature [34]. 

Coconuts are also often exposed to these extreme weather conditions throughout their entire productive lifespan [35]. Several studies have reported the direct impact of drought on nut yield, starting with inflorescence commencement and lasting until nut maturity. Co-occurring dry weather at critical stages of coconut growth, such as inflorescence primordium initiation, ovary development, and button size nut development, results in reduced nut output [36]. A prolonged dry weather would affect nut yield for a successive four years [37] and it is reported that seriously afflicted palms need six years to recuperate and produce enough nut yield [38]. Extreme temperatures of less than 10 °C or greater than 40 °C is detrimental to effective leaf area, nut set, and yield [37] contributing to reduced yield. 

Therefore, tackling the effects of global climate change on coconut must be given similar weight. Even though the essential chemical mechanisms behind these features are important, there is a paucity of research devoted to them, particularly in the case of coconut.

Major adaptive improvements are rarely overlooked by natural selection. The existing in situ coconut germplasm collections may contain genotypes that have naturally evolved to cope with heat and drought conditions since they have inhabited local environment long enough to be influenced by natural pressures [39]. As proposed by [13], historical climate in the environments where geo-referenced germplasm accessions were originally collected can be used to study local adaptability to abiotic factors. In addition, genomic tools in combination with environmental variables have been demonstrated to be useful for identifying adaptive trait loci and predicting phenotypic variation associated with abiotic stresses. For instance, genome-wide selection scans (GWSS) [23] and genome-environment associations (GEA) [24] have successfully been used to study natural adaptation of forest tree to heat stress [40] and common bean to heat [41] and drought stress [42], respectively.

Thus, similar approaches could be employed to effectively utilize the current coconut germplasm in identifying novel sources of genetic variation associated with their adaptation to changing climate. However, uncovering cryptic adaptive variation may be hampered by the fact that the abiotic stress tolerance is typically governed by multiple genes, each with minor effects [43]. Marker-assisted backcrossing (MAB), genomic prediction (GP), and genome editing are potential strategies for accelerating the genetic gain for this complex trait.

### 3.3. Pest and Disease Resistant Coconut

Severe damage caused by pests and diseases has been a major source of concern in various crop industries. Despite advances in chemical and biological management in agriculture, biotic infestation has had a substantial detrimental impact on crop productivity. Similarly, many biotic factors hinder coconut performance in various locations and has occasionally resulted in significant economic loss. 

Coconut mites (*Aceria guerreronis*), a small insect that lives beneath the perianth of nuts is a devastating pest that attacks growing nuts. It is reported that chemical control of these mites is less effective and unsustainable due to the high expense of recurrent applications [44,45]. As for biological control, there have been studies on the use of natural predators, however, their slow rate of multiplication and inefficient self-distribution in the natural environment remain constraints for widespread deployment [46]. Therefore, development of tolerant varieties/hybrids is necessary [26].

In addition to this, the Red Palm Weevil (RPW) *Rhynchophorus ferrugineus* is a devastating pest affecting the coconut industry, especially in South Asia [47]. In Malaysia, the first detection of RPW infestation was discovered by the Department of Agriculture, Malaysia in 2007 and the rapid spread of RPW was observed in 2011 [48]. Since the infestation starts with a buried tissue borer, RPW infestations are challenging to detect in the early stages. The coconut industry has seen major change as a result of RPW, particularly in Malaysia. However, since the first report in 2013, there has not been a comprehensive study to establish an effective chemical control mechanism for RPW or to develop resistant varieties/hybrids. According to a report, Malayan Yellow Dwarf (MYD) variety was found to be less preferred by the RPW to lay egg while the maximum number of eggs were laid in Chowghat Green Dwarf [49]. This suggests that some varieties may possess resistance/tolerance mechanism against this pest.

Coconut rhinoceros beetle (CRB) *Oryctes rhinoceros* L. is another serious pest of coconut. In 1909, it was inadvertently introduced from its native Southeast Asia to the island state of Samoa in the Pacific Ocean [50]. Since then, coconut growing areas in several countries and territories within the Pacific and India Oceans have been reported to be affected by this pest [51], most recently Vanuatu and New Caledonia in 2019 [52]. During the late 1960s and early 1970s, the discovery and distribution of a viral biocontrol agent, Oryctes rhinoceros nudivirus (OrNV), kept the CRB under control [3,4,5,6,7,8,9,10,11,12,13,14,15,16,17,18,19,20,21,22,23,24,25,26,27,28,29,30,31,32,33,34,35,36,37,38,39,40,41,42,43,44,45,46,47,48,49,50,51,52,53,54,55]. However, a novel CRB variant discovered through mitochondrial DNA sequencing has emerged, with increased resistance to the OrNV, and is suggested to be associated to the recent outbreaks [51]. With the advent of next-generation sequencing, comparative sequence analysis can be performed to identify OrNV strains that are more pathogenic to CRB. 

Lethal Yellowing Disease (LYD) is one of the major coconut diseases in the Caribbean and Latin America. More than thirty different varieties of palm trees, including coconut, are afflicted by this deadly disease [56]. Even though there is no cure for LYD, various measures have been used to reduce the spread such as quarantine, chemotherapy, vector control, sanitisation and cultivation of resistant varieties. Malayan Dwarf was used as one of the parents to produce F1 hybrids in a trial testing resistant varieties and was reported to have a significant level of resistance to LYD [57]. 

Root Wilt Disease (RWD) is another devastating phytoplasma disease which is transmitted by plant hoppers (*Proutista moesta*) and lace wing bugs (*Stephanitis typica*) and conforms to any other phytoplasma. This disease is difficult to identify as it cannot be cultured in vitro [58]. Integrated management has been suggested in previous studies as a mechanism to control RWD spread. [59] found that screening trials showed that Chowgat Green Dwarf had high resistance to RWD. 

Host plant resistance offers a long-term solution to addressing pest and disease-related losses in coconut [7]. To boost the coconut industry, efforts should be made to cultivate pest and disease resistant varieties and hybrids. Exploitation of genomic and transcriptomic studies is urgently needed to identify parents that can be used to produce hybrids resistant to the above-mentioned biotic problems that are severely damaging the coconut industry. Integrated management approaches and major replanting efforts with resistant varieties/hybrid are required to protect coconut industry.

## 4. Progress of Current Breeding Strategies and Potentials of Biotechnological Breeding Approaches and High-Throughput Genotyping Platforms

Coconut’s long generation time and perennial nature becomes a huge hindrance in its basic and applied research. As a result, there are significant limitations to genetic gain and the development of high-performing agronomic coconut cultivars [60]. Incorporating different biotechnological approaches such as marker-assisted breeding, genomic-assisted breeding, transcriptomics, and genome editing with speed breeding can accelerate the rate of genetic gain in a shorter breeding cycle, providing solutions for significant issues relating to coconut, such as plateaued yield and biotic stresses (Figure 1). The viability of current advances in biotechnological tools and key target traits to be manipulated in coconut will be discussed in the following section of the paper.

### 4.1. Molecular Markers and Marker-Assisted Breeding (MAB)

Since the early development of marker-assisted breeding (MAB), plant biotechnology has emerged as a key component of comprehensive research studies in various crops. MAB benefits plant breeding programs by reducing the time it takes to develop quality attributes, which could otherwise take more than ten years [44]. There has been considerable progress in the use of molecular markers as a tool for selecting and breeding desired traits in coconut and other major crops.

DNA fingerprinting which defines the unique molecular pattern of a genotype can be applied using various molecular markers such as codominant markers, microsatellite (SSR), Single Nucleotide Polymorphism (SNP), Randomly Amplified Polymorphic DNA (RAPD), Cleaved Amplified Polymorphic Sequence (CAPS), and Sequence Characterized Amplified Region (SCAR). In coconut, molecular marker was utilized to determine the palm’s height using RAPD [62]. Ref. [63] employed a pair of microsatellite primers that are more specific than SCAR primers to successfully differentiate Sri Lankan Tall from Sri Lankan Green Dwarf and Sri Lankan Yellow Dwarf. A SNP marker was developed to identify the aromatic and non-aromatic coconut, which differ only by a single point mutation [64,65]. Ref. [66] used EST-SSR markers to test the genetic purity of the progeny obtained from hybrids tall x dwarf. Since there was no evidence in identification of intra-varietal hybrid, i.e., dwarf and dwarf, [67] have conducted a study using DNA fingerprinting intended to verify the legitimacy of hybrid seeds of dwarf x dwarf crossings using the detected markers, aiming to promote intra-varietal hybrids production. 

### 4.2. Whole Genome Sequencing and Genomic-Assisted Breeding

Over the past several years, the cost of sequencing has continuously dropped, enabling more extensive studies of whole genome of important crops including coconut. A high-quality reference genome will facilitate the discovery of novel genes responsible for economically important traits such as pest and disease resistance, drought tolerance and high yield, among others, which consequently could drive genetic improvement of coconut. Also, the availability of full genome sequence can render a better understanding of the evolutionary history of coconut. In 2017, the first draft coconut genome was released from the cultivar Hainan Tall. A total of 419.67 gigabases (Gb) of clean reads were obtained by the Illumina HiSeq 2000 platform, representing 90.91% of the predicted *Cocos nucifera* genome (2.42 Gb). This study revealed that 72.5% of the coconut genome is composed of transposable elements, majority of which were long-terminal repeat retrotransposons (LTRs) (92.23%) [68].

A couple years later, the first whole genome sequence of a dwarf coconut, cv. Catigan Green Dwarf (CATD) was generated using PacBio Single Molecule, Real Time (SMRT) sequencing at 15× coverage of the 2.15 Gb expected genome size. After improvement using 50× Illumina paired end MiSeq and Chicago sequencing, the final assembly with 97.6% coverage of the estimated genome size was obtained. The data analysis resulted in a total of 34,958 protein-coding genes that are involved in several important traits, including such as resistance to pests and diseases, drought tolerance, and coconut oil biosynthesis [69].

More recently, Nanopore single-molecule sequencing and Hi-C technology were used to generate de novo assembly of two coconut individuals representing tall and dwarf varieties. A total of 29,897 and 28,111 gene models was annotated in the tall and dwarf genome, respectively. Notably, this study revealed the key genes involved in the divergence of tall and dwarf heights traits in coconut [70]. 

One of the main next-generation sequencing (NGS)-based approaches for SNP marker discovery is genotyping-by-sequencing (GBS) due to its adaptability, cost-effectiveness, and efficiency [71]. The identified SNP markers can be used for different purposes including genomic diversity studies, structural population studies, genetic linkage analysis, and genomic selection [71]. However, this technology has been uncommon in coconut, until recently when a group from Brazil reported the implementation of GBS using restriction-site associated DNA-sequencing (RAD-seq) to identify SNP markers for variability and genetic structure study of local dwarf coconut populations [72]. Another study also successfully utilized this approach to identify and develop SSR markers from 38 coconut accessions using Illumina GBS’s genomic sequencing data [73]. Apart from GBS and RAD-seq, other available high-throughput genotyping approaches are double-digest RAD (ddRAD) and restriction fragment sequencing (REST-seq) [74].

The reported studies on the sequencing of the entire coconut genome demonstrated that many novel genes and genetic markers have been discovered as well as some insights into the evolutionary history of the coconut. In the coming years, it is anticipated that the number of studies on the whole genome of coconuts will increase significantly, providing a large amount of genomic data that will facilitate future functional genomics and molecular breeding in this crop species. 

### 4.3. Transcriptome Sequencing

Transcriptomics has been intensively explored in many organisms and provides substantial insights into gene structure, expression, and regulation [75,76,77]. Advancements in transcriptomic technology have resulted from advances in sequencing technology [78,79]. RNA-Seq has been found to be more sensitive and have a higher throughput than traditional hybridization or microarray-based techniques for gene expression study [76]. As a result, novel elements of the transcriptional landscape of gene activity have been uncovered in a variety of organisms [80,81,82]. Several studies have utilized this technology to determine the genetic basis of various coconut traits.

The transcriptome profiling of coconut was obtained using Illumina RNA-Seq technology together with de novo assembly from a mixed tissue sample, which has substantially aided in providing basic information for molecular breeding and additional molecular biological study [60]. It is reported by the authors that 57,304 unigenes with average length of 752 bp were identified. It is also notable that, the attempt by [60] revealed 99.9% novel unigenes in relative to the released coconut expressed sequenced tag (EST) sequence. As a result, this work has contributed an invaluable resource for future research to isolate and characterize individual genes involved for several biochemical pathways in coconut.

Apart from that, Illumina paired-end sequencing was utilized to obtain gene expression pattern during somatic embryogenesis in coconut. The importance of somatic embryogenesis in coconut for bulk production of high-quality palms cannot be understated. Yet, the recalcitrant nature of coconut creates a bottleneck in somatic embryogenesis [83]. Understanding, identifying, and characterizing the molecular events during coconut somatic embryogenesis, according to the literature, will not only help establishing reproducible protocols, but also shed light on the complex relationship between plant growth regulators and various development stages during somatic embryogenesis. Besides, by using Illumina paired-end sequencing, ascertaining embryogenic potential of somatic cell is possible [83]. 

Transcriptomics serves an immense potential towards disease management strategies since it provides a lot of information about a sequence and a basic biological process [84]. The advent of transcriptomics via RNA-Seq has made possible to exploit the genome of various species [85,86] which have aided researchers and breeders in understanding the complexity of disease occurrence in plants. In India, for example, transcriptome profiling was used to compare healthy and afflicted Chowghat Green Dwarf coconut trees for the fatal root wilt disease. RNA-Seq detects similar patterns, allowing for in-depth research and fresh insights on the interactions of the palm with the pathogen that causes root wilt disease [87]. Another study by [88] discovered a key collection of genes associated with the coconut defence system in response to phytoplasma attack upon infection. This novel resistance and/or susceptibility-inducing genes can be utilized in breeding coconut for phytoplasma resistance. 

With its enormous seed and maintaining endosperm even at maturity, the coconut offers a unique opportunity to study epigenetic mechanisms involving both small RNAs and RNA-directed DNA methylation in developmental context [89]. To find probable homologs of components necessary for RNA-directed DNA methylation, de novo transcriptome assembly of RNA-seq libraries constructed utilizing seed tissues including mature embryos, gelatinous endosperm, and dwarf variety leaves was performed [89]. The study provides compelling evidence that RNA-directed DNA methylation and other small RNA-mediated silencing pathways are sustained and active especially in maturing endosperm. Similar evidence that RNA-directed DNA methylation plays a late role in endosperm formation was presented in an earlier work [90,91].

A study to understand coconut’s response to salt-induced stress provides interesting facts about the current population size of coconut and signaling pathways involved in salt stress response. The abrupt decline in population size is due to changes in ocean levels brought on by glaciations, which may have triggered the recent invasion of transposable elements into the genome of coconuts [92]. A plant’s initial response to salinity occurs within seconds to hours [93] and reactive oxygen species pathway (ROS) and oxidative stress signaling are both key mechanisms [94]. Ref. [92] have identified a gene encodes for chloroplastic CU-Zn superoxide dismutase (SODCP) on chromosome 8, which is found in Hainan Tall but not in the Aromatic Dwarf. The study suggested that expression of SODCP could be the marker for salt tolerance. As mentioned earlier in this paper, dwarf which has evidence of human domestication may have evolved into not containing the SODCP while Tall which remains in coastal region mostly sustains these characteristics.

In model plants i.e., rice and Arabidopsis, an Abscisic Acid (ABA) dependent pathway expressing protein phosphate 2C (PP2C) gene on chromosome 14 is upregulated when exposed to salinity stress [95], which is in contrast to coconut, whereby it is downregulated [92]. The physiological response of dwarf varieties towards salt stress is mediated by stomatal regulation. These findings are essential to enhance our understanding of preparing for climate change impacts by developing varieties adaptive to various soil types.

### 4.4. Clonal Propagation Via Somatic Embryogenesis

Currently, over half of the world’s cultivated coconut palms are senile and need to be replanted immediately to ensure adequate production to meet the ever-increasing demand for coconut products. In addition to ageing palms, biotic and abiotic stresses also affect coconut productivity. To address this issue, mass replanting using conventional means from the seed is a time-consuming and inefficient procedure to generate high quality and true-to-type planting materials. Alternatively, the rapid large-scale production of the new high-quality and disease-free coconut palms can potentially be achieved using in vitro culture. 

Several somatic tissues including young leaves, young stems and rachillae of young inflorescences were used as starting materials to form embryogenic calli in the early attempts of coconut somatic embryogenesis (SE) [96,97,98]. Later, clonal plantlets from zygotic tissues such as embryos and plumules have been obtained, however with a low success rate [99,100,101]. More recently, different explants have been used to induce callus such as inflorescence, leaf, unfertilized ovaries, immature embryos, and endosperm, but plumular tissues appeared to be the most suitable for SE [102,103,104,105,106,107,108]. 

It is well known that coconut is among the most recalcitrant species to in vitro regeneration, and it remains a major bottleneck in coconut research. Among the reasons include the heterogeneous response of various coconut explants, the genotype-dependent regeneration efficiency, and the slow growth of regenerated plants under in vitro as well as ex vitro conditions [109]. For SE, various factors have been found to play an important role in its success including the genotype of the donor palm, explant type and age, media composition, plant growth regulators (PGR) concentrations, and the acclimatization procedures subjected to the regenerated plantlets.

With the recent development of novel molecular techniques, further improvement of SE process in coconut can be made through a better understanding in the underlying mechanism and identification of key genes related to the embryogenic competence. In recent studies, several genes and micro RNAs (miRNAs) involved during different SE stages in coconut have been identified using genomic and transcriptomic sequencing approaches [110,111]. This valuable information will be useful for future studies to improve the current protocols. 

### 4.5. Genome Editing

The advent of genome editing tools has revolutionized and accelerated the plant research over the last decade. Meganucleases, zinc finger nucleases (ZFNs), and transcriptional activator-like effector nucleases (TALENs) are dubbed as first-generation of genome engineering nucleases, while clustered regularly interspaced short palindromic repeat/CRISPR-associated system has gained more attention as the second generation due to its simplicity, efficiency and cost-effective. These genome editing tools share the same underlying principle by generating DNA double-strand breaks (DSBs) at a targeted genomic location which then being repaired either via non-homologous end joining (NHEJ) or homology-directed repair (HDR) by plant endogenous repair mechanism [112]. The generation of targeted DNA changes such as substitution and insertion/deletion (indels) may result in gene function alteration, which then lead to modification of external features (traits). For instance, targeted mutations can be induced using CRISPR in susceptible coconut to confer resistance to major diseases. Some of the genes that can be targeted include the NBS-LRR domain, PR1, PR4, pathogenesis-related genes transcriptional activator PT15-like gene, thaumatin-like protein, HSP70 and glutathione S-transferase, which are associated with root wilt disease (RWD) [113].

Despite of its wide application in other crops, the genetic improvement of coconut via this approach remains challenging. To date, no genome editing tools have been successfully applied and reported for coconut. This little progress is due to the lack of established tissue culture and genetic transformation methodologies. At present, there is only a single report on the transient genetic transformation of embryogenic callus of coconut [114]. Ideally, the functional characterization of genes and elucidation of metabolic pathways involved in biotic and abiotic stress regulation should be done in a stable transformation of coconut. Notwithstanding, it will require lengthy time to observe the traits of interest owing to its long-life cycle. To address this, previous studies used in vitro biochemical test [115,116] and transformation of gene in model plants such as Arabidopsis thaliana and rice [117,118]. Since the success of genome editing technology in crops is mostly dependent on well-established transformation methods, more studies are required to develop an amenable and efficient genetic transformation of coconut tissues.

## 5. Speed Breeding to Accelerate Coconut Breeding

Speed breeding is a promising innovative technology for accelerating genetic gain by manipulating the growth environment of plants [119]. This technique involves growing plants under constant light for an extended period (20–22 h), allowing them to photosynthesize longer, resulting in faster growth. By increasing the number of generations per year, researchers can reduce the time required to develop new varieties [120]. The manipulation of plant growth conditions can be done either using a DIY benchtop cabinet, growth chambers, or glasshouses, depending on the available resources and crop type [121]. Speed breeding has been successfully applied to several crop species, including wheat [122,123], barley [124], oats [125], canola [126], among others. 

Implementing a speed breeding algorithm in coconut breeding would be beneficial due to the long generation cycle of coconut. Coconuts undergo a lengthy juvenile period before entering their reproductive stage (Figure 2). As reported in a study by [127], reducing the juvenile stage of coconut must be the prime objective in coconut breeding program.

A saturated photosynthetic photon flux density of about 1400 µmoles photons m^−2^s^−1^ indicates a typical C3 plant nature of coconut [128]. Slow growth of coconuts is observed after the sprouting phase, during the 12th and 14th month, as photosynthetic rates gradually decline due to the self-shading effect of upper canopy fronds. Coconut becomes a resource-constrained crop as the amount of photosynthetically active radiation (PAR) and leaf nitrogen decreases. As a result, coconut is found to be less efficient than other C3 plants at converting PAR energy into biomass [129]. The highest energy conversion of PAR into dry matter has been estimated to be 1.2–1.4 g MJ^−1^.

On the contrary, protocols on growing conditions such as soil media composition, lighting, and temperature are not available to accelerate the growth of palm. Although parameters for bread wheat and durum wheat (C3 plants) could be adapted for coconut, more research on the spacing and coordination of sprouting coconut is necessary to assess whether the benefits of speed breeding outweigh the costs. Apart from this, according to [128], prolonged exposure to intense light could lead to seedling death. Therefore, extra caution should be taken to avoid overexposing the coconut seedling to high light exposure which could be lethal.

## 6. Next Generation Phenotyping Approaches

Phenotype information is the link between an organism’s genetic make-up (genotype) and environmental factors. A genotype-phenotype map (G-P map) of coconut would enable breeders to understand the dynamics of the entirety of the palm’s lifespan. The G-P map is the result of extremely complex dynamics that include environmental factors, and understanding these dynamics is synonymous with bridging the genotype-phenotype gap [130].

The morphological study of coconut palms, particularly the tall variety, is difficult. On the other hand, phenotyping the dwarf variety, which will eventually grow taller, would pose a similar problem. Studying yield-contributing traits like number of bunches/palm/years, number of fruits/bunches, stem height, petiole length, and overall frond length is not only time-consuming, but it can also be destructive. Nevertheless, the documentation of morphological traits is crucial as it will benefit farmers and breeders in future crop selection and improvement. Therefore, fast, accurate, and high-throughput methods need to be developed for collecting the phenotyping data.

The current state of the global coconut industry necessitates an increase in yield to meet rising demand and ensure food security. Even though other biotic and abiotic stresses are equally important, the focus of this paper will be on a method for morphological characterization of traits that contribute to yield. The novel approach described here could serve as a springboard for the development of phenotyping for other traits. 

First, a clear understanding of morphological traits that contribute directly to yield and inheritability must be identified for a successful application of next generation phenotyping (NGP). Other studies have found that coconut leaves make a significant contribution to the yield. In comparison to inflorescence and fruit traits, leaf characters are more stable [131]. A detailed analysis of genotypic correlation coefficients, which is partitioned into direct and indirect effects through path-coefficient analysis to identify the true contribution of morphological traits to yield, reveals that the number of functional leaves, followed by petiole length, and leaf length, exerts the greatest direct effect on yield [132].

The Standardized Research Manual in Coconut Breeding (STANTECH) lists the length of the petiole, petiole color, length of the whole frond, and measurements of the width and length of four middle leaflets from the 14th frond as the minimum measurements needed for leaf characterization of coconut. However, sampling the wide range of germplasm that is available in producer countries requires a lot of labor and can be destructive as the whole 14th frond must be removed at the point where the petiole connects with the stem. The above-mentioned measurements are necessary for leaf characterization in palms such as coconut and palm oil, to estimate the total area of leaves/canopy of individual palms. Thus, measurements of the width and thickness of the petiole are employed to calculate cross sectional area, and indirect estimations of the dry weight of the entire leaf are obtained [133]. 

A non-destructive method for estimating the leaf area of oil palms was developed [134], in which a similar method has been adapted for coconut using the STANTECH manual. This method can be further improved by adapting new tools and software for the data collection process. “Plant Screen Mobile” (PMS), a mobile app that can be used on an Android phone, has the potential to make phenotyping coconut leafiness easier, faster, and more precise than the traditional method. This app allows researchers to estimate projected leaf area, color, and shape parameters in a variety of plant architectures [135].

By integrating the PMS with the UAV, the number of bunches and fruits as well as observed crown appearances will be easier to measure, particularly in tall varieties (Figure 3). As a result, this pipeline for generating phenotyping data may not only improve data accuracy, but also reduce the time and cost for data collection.

DotDotGoose (https://biodiversityinformatics.amnh.org/open_source/dotdotgoose/) (accessed on 14 March 2020) is another open-source program for manually counting objects from an image. This program can be employed to manually count the number of palms in a population using aerial images captured by a UAV. With increased sensitivity, the counting of individual fruits and bunches will be feasible and useful for tall varieties.

For optimal results, existing tools and software must be adapted for use in coconut research. Continuous advancements in the automation of these tools will improve the efficacy of coconut selection and future research. Coconut is a rain-fed perennial palm with adventitious roots, a condition in which root growth could be both normal and as a result of external suppressing conditions. Due to impending extreme climate change, plantation of perennial crops requires advanced development to acquire tolerance to abiotic stresses, particularly drought and high temperature tolerant varieties [136]. It is critical to identify genotypes/varieties with highly effective root development that can withstand harsh climates. It is estimated that long dry periods reduce productivity by 50%, therefore improving tolerance for adverse climatic conditions will boost coconut productivity [137].

Since natural selection has tested more possibilities than humans would ever examine, spontaneous variation can be an important source of resistance to various abiotic stresses [138]. The discovery of genotypes/varieties in the existing in situ coconut germplasm collections that display local adaptation to a changing environment would allow the genetic dissection of complex adaptive traits for pre-breeding and climate adaptation. However, quantitative traits such as drought and high temperature tolerance are often regulated by many genes with small effects, making their research challenging [43]. 

Recent advances in predictive breeding approaches, such as genomic prediction (GP) [139] provide valuable information on the frequencies of different alleles of polygenic traits for adaptation to climate change, speeding up genotypic selection from natural sources of current germplasm [140].

A phenotypic and physiological parameter for selecting drought tolerant traits among coconut is demonstrated via simulation modelling from calibrated and validated data from previous studies on various perennial crops in India [141]. Similar work has also been pursued and reliable parameters such as leaf stomatal frequency, stomatal index, chlorophyll fluorescence, epicuticular wax content, and lipase and protease activity were suggested for physiological assessment for drought tolerance in coconut [142,143]. Cultivars with finest density with more roots appeared to be less affected by drought [144]. Besides this, analyses and reports indicate that tall varieties or hybrids with tall mothers are more tolerant to drought conditions than dwarf varieties or hybrids with dwarf mothers [145,146,147].

Most of the previous studies relied on manual observation to obtain quantitative data such as plant height, leaf color, chlorophyll content, disease sensitivity, yield, biomass, and drought tolerance, which is not only time consuming but also labor-intensive. The biological nature of the coconut including high stature, long generation time and low multiplication rates make the large-scale screening difficult and destructive. As a result, coconut phenotyping has lagged behind genotyping, contributing to the slow progress of fundamental research. As reported in European Plant Phenotyping Network [148], the challenge in the quantitative analysis of crops is already a bottleneck.

Thus, advanced phenotyping protocols are required to understand the relationship between environmental factors, genotypes, and phenotypes in order to enhance coconut performance [149]. To address the “phenotyping bottleneck”, the development of high-throughput, cost-effective and accurate methods such as autonomous ground vehicles/rovers (AGVs), unmanned aerial vehicles (UAVs), and spectral satellite imaging [150] is essential. 

The combination of fundamental biology, data sciences, and engineering, among others to obtain multidimensional phenotypic data of an organism as a whole is known as phenomics [151]. This high-throughput phenotyping uses robotics, high-tech sensors, imaging systems, and computing power to screen a large population in the field and connect the results to available genomics data for analyzing coconut performance. 

There are various methods that can be utilized to obtain phenotypic data from coconut. Infrared (IR) and hyperspectral imaging are two of the easiest and most cost-effective ways to identify drought and high temperature tolerant cultivars. These methods are based on recent experiment by NASA using satellite platforms equipped with multi and hyperspectral sensors, Ecosystem Spaceborne Thermal Radiometer Experiment on Space station (ECOSTRESS), Soil Moisture Active Passive (SMAP) and Hyperspectral InfraRed Imager (HyspIRI), which collect high-resolution spectral and environmental data providing applications for water use efficiency [152,153,154]. The innovative technology is available in various forms and applications such as IR thermometer which use digital reading, field-mounted IR sensors, handheld/field-mounted IR cameras and IR cameras fitted in UAVs. IR and hyperspectral imaging are based on the idea of canopy temperature differences which will be sensed through the release of IR energy and then converted into electrical signals by imaging sensors. This signal will be shown as monochrome or colored data on a computer screen. 

Stomatal conductance and canopy temperature are surrogate measurement tools whereby a cool canopy indicates a high transpiration rate of the palm while a high temperature denotes a lower transpiration rate [136]. This allows for the rapid and non-destructive identification of varieties with the most efficient root system against drought and high temperatures. If the IR and hyperspectral studies are conducted in high temperature, drought prone, and hot weather areas where coconuts are grown, there is a high chance that data and varieties for drought and high temperature tolerant coconut cultivars will be obtained.

## 7. “Grower’s Experience” vs. Deep Machine Learning

Agriculture is a one-of-a-kind industry with economic, biotic, and abiotic influences. Plants and soil have long been the only inquisitors as well as sensors available for quantitative and qualitative characterization. These manual methods are widely utilized to guide operational decisions, irrigation management, pest and disease prediction, monitoring, and ideotype selection. However, manual interventions in the hope of improving the yield and quality of a crop are less reliable, laborious, and prone to error. In the coconut domain, previously used quantitative and qualitative characterization methods and tools were ineffective for current and future challenges. The coconut’s heterozygosity, monotypic, perennial nature, and long generation cycle are significant deterrents. On the other hand, the experimentation process necessitates a large amount of land. Furthermore, because there are no known wild or domesticated related species of coconut, the genetic pool sources for extracting useful relationships and patterns in collection, selection, and breeding process are absent.

The transformation of traditional quantitative and qualitative tasks is required as coconut cultivation intensifies. Data mining, which includes machine and human-generated data then uses the Internet of Things (IoT) as a booster, is a channel for combining findings from previous biotechnological studies, phenotype and genotype data from coconut cultivation improvement efforts, and phenotype and genotype data from farm output expansion and diversion (Figure 4).

The fields of genetics and genomics have seen progress with the application of machine learning (ML), such as for ecological niche modelling [155] and functional genomics [156]. This powerful tool has resolved the issue of handling and making sense of enormous genomic datasets, which has opened up many possibilities in population and evolutionary genetics.

ML-based phenotyping is feasible to accelerate the rate of genetic gain in crops and the throughput of crop screening, which facilitates the early detection of diseases for a reduction in crop yield loses [157]. Thus, ML is a profound tool to replace traditional ways of detecting major problems that have been haunting the coconut industry, such as irrigation management, yield prediction, and pest and disease prediction. In the context of abiotic stress tolerance, the ability of ML to combine heterogenous datasets from different sources (e.g., genotype, phenotype, climate data) while avoiding the curse of dimensionality may help improve the prediction of polygenic local adaptation which is often influenced by environmental variables [13]. By incorporating the genomic analytical tools such as GWAS, GWSS, and GEA with ML, this would enable the breeders to select parents more accurately to be used in coconut pre-breeding and breeding program.

Integrating artificial intelligence in data mining for coconut cultivation and breeding regimes could result in considerable progress in applying smart farming to rapidly revive the coconut industry. However, proper management of data and sharing protocols must be developed so that farmers and researchers can have a quick access to the data and gain profound effects. Farmers’ accessibility in performing smart farming must be studied.

## 8. Concluding Remarks

The application of individual tool or technology in isolation would hamper crop improvement. Thus, to attain a profound effect in farm output, integration of novel biotechnological and precision agriculture tools with selected conventional methods in breeding pipeline is crucial. To accelerate coconut genetic improvement, genomic-assisted breeding, transcriptomics, next-generation sequencing (NGS)-based genotyping, genome editing, precision phenotyping and speed breeding, among others need to be integrated which could facilitate breeding and selection decisions in breeding program. 

The high-throughput and cost-effective marker systems derived from NGS-based platforms will enable rapid genotyping process. As the sequencing cost is expected to continuously drop, more coconut genotypes can be re-sequenced to construct coconut “pan genome” which would provide novel insights into the genetic basis of complex traits such as yield and drought tolerance. At the same time, considerable efforts should be made to train more experts on handling the “big data” generated from the NGS platform as well as enhancing the computing capability. The genetic dissection of resistance mechanism towards important pests and diseases in coconut can be further elucidated with the increased knowledge at genomic and transcriptomic levels. With a growing number of crops being successfully optimized for speed breeding, similar achievement in coconut will become a game changer in coconut breeding pipeline. Field data collection especially related to yield trait might be seen as a major constraint due to the biological nature of coconut. The proposed method using image capture app coupled with UAV in this paper could potentially address this issue and serve as a catalyst for further development of high-throughput approaches. 

Prioritizing the utilization of different tools in parallel with the objectives of the breeding program is vital to resurrect the coconut industry with current increasing demand as well as a preparation for future challenges. Molecular markers, precision phenotyping tools such as UAVs, and spectral satellite imaging are readily available to be used for identification of high-performing palms in the germplasm collections. As genomic resources are expected to further expand in the coming years, biotechnological tools such as NGS-based genotyping, whole genome sequencing, transcriptome sequencing, and pangenome analysis can be applied in the medium-term to advance coconut breeding. The application of clonal propagation and genome editing may take a while as further research work is required to optimize the tissue culture protocol of coconut. Although it is still early days, success in adopting speed breeding technology to shorten the generation time of coconut will be a breakthrough in coconut research.

## 9. Future Directions

The potential of natural variation as valuable sources of resistance to biotic and abiotic stresses has been discussed in this review through the study of local adaptation in germplasm collections assisted by GWAS, GWSS, and GEA using available phenotypic, genotypic, and climate data. To speed up coconut breeding cycles, marker-assisted selection and marker-assisted backcrossing provide an alternative to traditional breeding for the introgression of target genetic variants from exotic germplasm into elite cultivars. However, these approaches have proven to be useful for traits which are governed by a few genes such as resistance to biotic stresses but may be inefficient for tracing quantitative traits which are often regulated by many genes with small effects such as tolerance to abiotic stresses. In the near future, genomic-assisted backcrossing (GABC) may become more common to facilitate the introgression of polygenic traits into the cultivated coconut genepools. Another exciting tool that can be employed for this purpose is genomic prediction (GP), which promises to expedite selection and pre-breeding for complex polygenic adaptive traits from natural sources. We envision that this predictive breeding method will also help future genebank characterization to mine the molecular signature of selection and adaptation of coconut for climate adaptation.

## Figures and Tables

**Figure 1 plants-11-03414-f001:**
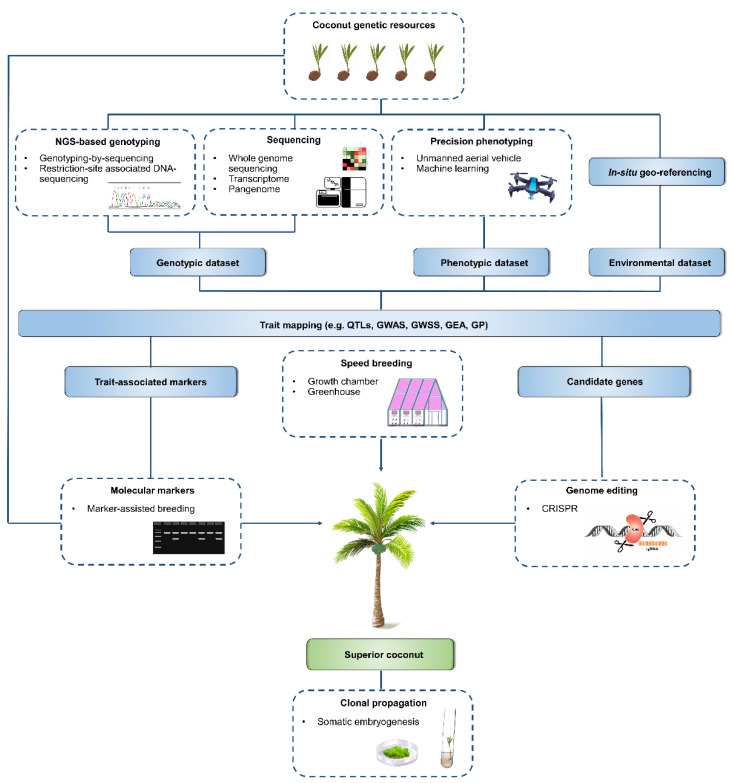
Utilizing and integrating various modern breeding technologies for genetic improvement in coconut. Coconut genetic resources such as conserved germplasm serve as a valuable source of new genetic variation. Next-generation sequencing-based genotyping approaches generate markers for genome analysis at a reduced complexity level. Next-generation sequencing platforms generate high-quality reference genomes and facilitate transcriptome as well as pangenomic analyses. Precision phenotyping expedites acquisition of phenotyping data of a large population in the field. The integration of these three approaches provides a powerful means to link genetic variations with the relevant phenotypes, allowing trait mapping using various approaches such as quantitative trait loci (QTL) and genome-wide association studies (GWAS). In situ georeferencing from genotypically characterized germplasm generates environmental dataset. The combination of genomic tools with environmental variables allows identification of adaptive trait loci associated with abiotic stresses via genome-wide selection scans (GWSS), genome-environment associations (GEA) [61], and genomic prediction (GP). The identification of trait-associated markers will facilitate the selection of individual palms based on their genotype in the breeding program. The molecular markers can be also utilized for characterization and evaluation of coconut germplasm. Once the candidate genes associated with the desired trait are identified, functional characterization can be performed using genome editing (e.g., using CRISPR) which enables a precise and accurate modification of targeted genes. Speed breeding will accelerate the entire coconut breeding progress. Clonal propagation allows rapid large-scale production of the new high-quality, abiotic stress tolerant, and biotic stress resistant coconut palms.

**Figure 2 plants-11-03414-f002:**
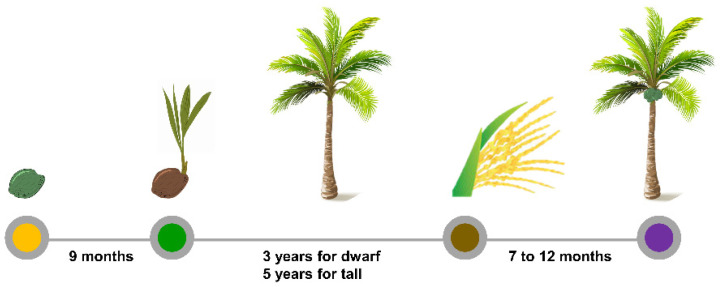
Life cycle of coconut palm. The duration of nut to sprout takes around 9 months, followed by the rapid formation of the haustorium and development of the root within several weeks. Then, at the juvenile stage (from sprouting stage to the first flower production), the growth in the tall variety is slower (5 years) and shorter in the dwarf (3 years). The flowers (male and female) are produced on a monthly basis at the spathe. In the second month of fertilization, the inner cavity will begin to differentiate. The fruit will take 7–12 months to reach its maximum size.

**Figure 3 plants-11-03414-f003:**
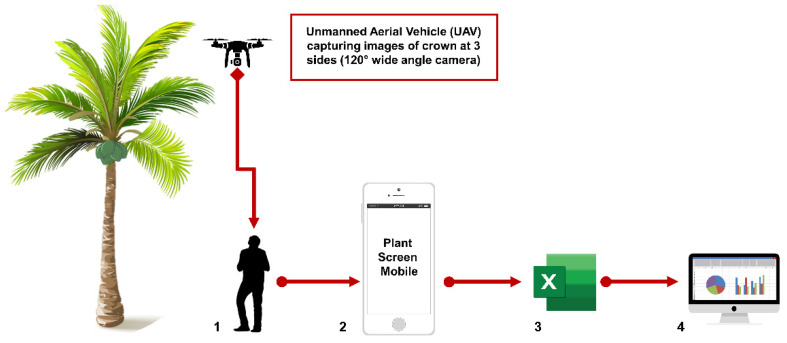
(1) UAV is utilized to capture the image of crown at three sides (for camera with 120° wide angle). (2) The image is transferred into the Plant Screen Mobile App (PMS) folder. (3) The total leaf area is automatically generated in CSV format. (4) Data analysis with relevant tools to obtain desired observations.

**Figure 4 plants-11-03414-f004:**
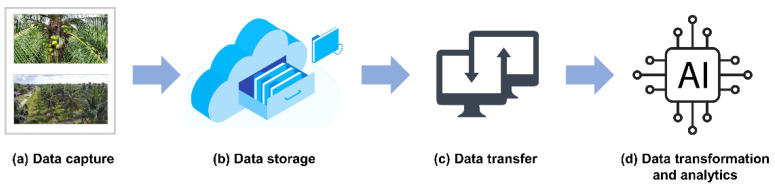
Agricultural data intervention to correct the performance of the farm processors. (**a**) Initial stage of data intervention, data capture is acquisition of data collected from sensors, UAVs, open data, genotype information, biometric sensing, and reciprocal data. (**b**) Data storage phase uses cloud-based platform. (**c**) Data transfer phase integrates cloud, wireless and linked open data network. (**d**) Data transformation and analytics utilizes machine learning platforms which will create automation in selecting, breeding and farm practices.

## Data Availability

Not applicable.

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
