# Peer review of "Improving Coconut Using Modern Breeding Technologies: Challenges and Opportunities"

_plants, 2022, doi:10.3390/plants11243414_

Round 1

Reviewer 1 Report

The work by Arumugam and Asyraf provides indeed a very thoughtful review addressing the perspectives of modern pre-breeding and breeding in coconut. The work is well written, and highlights key findings in the field. The literature compilation is substantial and offers new perspectives to the readers.

However, ss detailed below, my major suggestions are to (1) expand the main figure 1 in order to sketch the overall integration of novel resources for coconut with a hypothetical pre-breeding scheme, and to (2) follow a PRISMA procedure (http://www.prisma-statement.org/PRISMAStatement/FlowDiagram) at Materials and Methods in order to make the review more systematic. Before recommending acceptance, I would also suggest authors to address the following improvements.

First, please list explicit review goals at the end of the introduction section (L59) and a priori hypotheses to be explored by the review (L62). This will allow readers focusing on concrete questions and expected trends on how to effectively utilize novel coconut resources and variation.

Second, when reporting literature compilation, I encourage authors to follow PRISMA’s guidelines to prepare reviews. Please include a brief specific methodological section (after L66) explaining the concrete steps/parameters used during the search (i.e. keywords, search equation, target repositories), filtering, and summary of key references. This section can be brief (one paragraph), but yet would make the literature search for novel coconut breeding strategies more repeatable. I bet carrying out a more systematic search be condensed in a new table 1 with key studies.

Third, the review is missing a key mention that broadly summarizes the perspective on how to effectively utilize novel genetic resources within pre-breeding schemes. The main figures in Genes 2021 12:783 and Genes 2022 13:1 may serve as guidance.

When commenting on the utility of machine learning techniques (L616), I invite authors including seminal works that have innovated machine learning techniques to leverage genetic resources (i.e. Nat Rev Genet 2015 16(6):321-32, Trends Genet 2018 34(4):301-12). For instance, convolutional neural network, deep belief network (briefly mentioned in L616), multivariate Poisson deep learning, multilayer perceptron, probabilistic neural network or radial basis function neural networks may help improving the prediction of polygenic local adaptation by integrating heterogeneous datasets while side stepping the curse of dimensionality (Front Plant Sci 2020 11:580136). Please couple with my second comment above, specifically by expanding accordingly the keywords and search equation.

More specifically, I recognize authors for mentioning thoughtfully throughout the review the lack of crop wild relatives for coconut as natural reservoirs of standing adaptation to abiotic stresses (L627). Still, I am missing key examples supporting the utility of naturally adapted genotypes (refer to Front Plant Sci 2020 11:583323 and Plants 2021 10:2022”) to effectively source cryptic adaptive variation (see Front Genet 2022 13: 910386) in terms of specific abiotic stress such as heat (e.g. Front Genet 2019 10:954, Front Genet 2020 11:564515, and Genes 2021 12:556), and drought (e.g. Front Plant Sci 2018 9:128, and PLoS One 2013 8(5):e62898). Ultimately, what are the changes to simultaneously pre-breed for tolerance to various abiotic stresses? (to be expanded in L155 and L558)

As another minor point, micro-scale patterns are also overall overlooked; despite they have served as local refugia for in situ diversity and conservation (e.g. Ecosphere 2019 10:e0276, Catena 2020 193:104626, and Science 2020 368, 772–775), and are likely to have shaped within species adaptive potential. I encourage authors to discuss local-scale processes more thoroughly, and propose novel methodologies (such as gap analysis and down-scaling; please check Divers Distrib 26 2020 730-742, and PLoS One 5 2010 e13497) aiming to address microhabitat effects. 

Last but not least, in order to improve readability, I would recommend splitting the last section (L654) into individual “Concluding remarks” and “Future Directions” sections. The latter should start by discussing the potential caveats of other studies until now to effectively utilize natural variation within existing breeding programs, and propose new avenues of research. Authors must envision novel strategies to assist in the near future further genebank characterization and introgressive breeding into the cultivated genepools (e.g. via genomic-assisted backcrossing as a replacement to more traditional marker-assisted backcrossing. In the latter technique, please also refer to the following examples in which cryptic resources have successfully been coupled with backcrossing schemes to pre-breed respectively for abiotic tolerance, agronomic performance, biotic resistance, and nutrient content: Crop Science 2003 44:637-645, Theoretical and Applied Genetics 2006 112:1149-1163, Crop Science 2008 48:562-570, and Theor Appl Genet 2012 125: 1015-1031). Mind the potential of predictive breeding (i.e. genomic selection, briefly mentioned in L648, e.g. G3 2016 6:1819-1834) to select and pre-breed for complex polygenic adaptive traits. This way the review will have a broader scope, being more interesting to many more readers of Plants. 

Finally, please improve the resolution of figure 1 and 4.

Author Response

Response to Reviewer 1 comments

Point 1: However, as detailed below, my major suggestions are to (1) expand the main figure 1 in order to sketch the overall integration of novel resources for coconut with a hypothetical pre-breeding scheme,

Response 1: We thank the reviewer for this suggestion for improving the figure 1. We have re-arranged the various modern breeding technologies and added some steps of pre-breeding and breeding scheme.

Point 2: and to (2) follow a PRISMA procedure (http://www.prisma-statement.org/PRISMAStatement/FlowDiagram) at Materials and Methods in order to make the review more systematic.

Response 2: We thank the reviewer for this suggestion to make the review more systematic. However, we have sent this manuscript to be considered as a review article, not as a systematic review. Therefore, we have maintained the structure and format as is.

Point 3: Please list explicit review goals at the end of the introduction section (L59) and a priori hypotheses to be explored by the review (L62). This will allow readers focusing on concrete questions and expected trends on how to effectively utilize novel coconut resources and variation.

Response 3: Good point. We have added a few lines of text describing the review goals and priori hypotheses at the end of the introduction section.

Point 4: When reporting literature compilation, I encourage authors to follow PRISMA’s guidelines to prepare reviews. Please include a brief specific methodological section (after L66) explaining the concrete steps/parameters used during the search (i.e. keywords, search equation, target repositories), filtering, and summary of key references. This section can be brief (one paragraph), but yet would make the literature search for novel coconut breeding strategies more repeatable. I bet carrying out a more systematic search be condensed in a new table 1 with key studies.

Response 4: Please see response #2.

Point 5: The review is missing a key mention that broadly summarizes the perspective on how to effectively utilize novel genetic resources within pre-breeding schemes. The main figures in Genes 2021 12:783 and Genes 2022 13:1 may serve as guidance.

Response 5: We thank the reviewer for this suggestion for improving the manuscript. We have added a few lines briefly discussing the various recent approaches that can be used to effectively utilize novel genetic resources within pre-breeding schemes at the end of section 2: Collection, conservation, and utilization of coconut genetic resources. 

Point 6: When commenting on the utility of machine learning techniques (L616), I invite authors including seminal works that have innovated machine learning techniques to leverage genetic resources (i.e. Nat Rev Genet 2015 16(6):321-32, Trends Genet 2018 34(4):301-12). For instance, convolutional neural network, deep belief network (briefly mentioned in L616), multivariate Poisson deep learning, multilayer perceptron, probabilistic neural network or radial basis function neural networks may help improving the prediction of polygenic local adaptation by integrating heterogeneous datasets while side stepping the curse of dimensionality (Front Plant Sci 2020 11:580136).

Response 6: We have added a few lines discussing the application of machine learning in genomics study and how it can improve the prediction of polygenic naturally adaptive traits at L723 in the revised manuscript.

Reviewer 2 Report

General comments:

This is a comprehensive review of the current status and challenges for coconut breeding and the (potential) solutions offered by new breeding technologies. The manuscript is well written and will be of interest to coconut breeders and researchers especially and more generally to breeders and researchers involved in the genetic improvement of tree crops. The different types of modern breeding tools that could be used to improve or accelerate coconut breeding are also well described.

Given that (very) few of these tools are actually being used, it would be of interest to the readership to include a 'priority setting', i.e. what specific new  technology could be applied in the short, medium or long term to advance coconut breeding. 

Specific comments

Overall, the manuscript is very well written. however, here and there sentences are incomplete or unclear. e.g.

294-295: coding genes that are involved in several important traits [60].

Specify the traits.

377-378: Yang et al. [83] have identified chromosome 8 which codes for chloroplastic CU-Zn superoxide dismutase (SODCP) which is found in Hainan Tall but not in the Aromatic Dwarf.

Rephrase ‘have identified gene...... on chromosome 8 .... ‘

563-564: Phenotypic and palm physiological studies are screening process of germplasm evaluation for drought resistant coconut palm, which takes longer time

Unclear sentence

577: which is not only laborious but time consuming

check sentence

610-611: This enables the identification of the variety with the most efficient root against drought and high temperatures without any destruction and in a shorter amount of time.

Sentence unclear

Author Response

Response to Reviewer 2 comments

Point 1: Given that (very) few of these tools are actually being used, it would be of interest to the readership to include a 'priority setting', i.e. what specific new  technology could be applied in the short, medium or long term to advance coconut breeding.

Response 1: We thank the reviewer for this suggestion for improving the manuscript. We have now mentioned the specific new technology that could be applied for short, medium, and long term coconut breeding at the end of concluding remarks section.

Point 2: Overall, the manuscript is very well written. however, here and there sentences are incomplete or unclear. e.g.

Response 2: We thank the reviewer for raising this concern to improve some of the sentences. We have now corrected them.

Round 2

Reviewer 1 Report

Great improvements and wonderful artwork. My last recommendation is to better support the GEA approach referred to in Figure 1, L303 and L737. Specifically, please include (L303 would be a good place) the recent recommendations made by López-Hernández (Front Genet 2022 13:910386, doi:10.3389/fgene.2022.910386). In this sense, Figure 1 must be aligned with Figure 1 in that review. Also, please better exemplify how this approach could be extended in coconut research for heat tolerance (presumably key stress for the species) by referring to (e.g. L130) the case study on GEA for heat tolerance (Front Genet 2019 10:954, doi: 10.3389/fgene.2019.00954)    

Author Response

Response to Reviewer 1 comments

Point 1: My last recommendation is to better support the GEA approach referred to in Figure 1, L303 and L737. Specifically, please include (L303 would be a good place) the recent recommendations made by López-Hernández (Front Genet 2022 13:910386, doi:10.3389/fgene.2022.910386). In this sense, Figure 1 must be aligned with Figure 1 in that review.

Response 1: We thank the reviewer for this recommendation for improving the figure 1. We have added some steps in the figure 1 and a few sentences at L303 to support the GEA approach.

Point 2: Also, please better exemplify how this approach could be extended in coconut research for heat tolerance (presumably key stress for the species) by referring to (e.g. L130) the case study on GEA for heat tolerance (Front Genet 2019 10:954, doi: 10.3389/fgene.2019.00954)   

Response 2: We thank the reviewer for this suggestion to improve the manuscript. We have added the suggested paper at L216 to support the point.